# OpenReview forum: "CursorCore: Assist Programming through Aligning Anything"
_ICLR.cc/2025/Conference — Submitted to ICLR 2025_

### Official Review · Reviewer_zutF · 2024-10-25

**Soundness:** 3
**Presentation:** 2
**Contribution:** 2
**Rating:** 5
**Confidence:** 3

**Summary:**

This paper presents (1) a new conversational framework called Assistant-Conversation aimed at simplifying the programming process.  The paper also presents (2) a benchmark set for assisted programming, namely APEval, which has 164 benchmarks that are hand-curated starting with HumanEval and adding some or all of the "history", "code", and "user" components.  Since generating APEval like benchmarks is time consuming manual process, the paper also presents (3) a method, Programming-Instruct, to automate data collection.  Finally, the dataset is used (4) to fine-tune LLMs and get CursorCore models.

Assistant-Conversation framework: The framework consists of the elements system (S), history (H), current snapshot of code (C), user query (U) and assistant response (A).  The paper observes that current models have limitations in dealing with these 5 pieces.  This point about history (H) not being incorporated in prediction has been made previously in many papers in software engineering conferences; see, for example, see the FSE 2023 paper titled "Grace: Language models meet code edits."

APEval: APEval is an extension of HumanEval generated by asking programmers with varying levels of experience to annotate HumanEval by interacting with an LLM. The paper identifies four categories of benchmarks, where we either have H,C,U or just H,C, or just C, U or just C. Benchmarks for each category are generated by humans starting from HumanEval benchmarks.  There are a total of 164 benchmarks in APEval, 41 of each category, where history (H) ranges from 21-139 lines, code (C) ranges from 8-31 lines, and user query (U) ranges from 3-19 lines (Table 1).  The benchmark set could be useful contribution, but there is no mention of releasing the benchmark set in the paper.

Programming-Instruct: The idea here is to get a history of code snapshots generated in the process of solving a programming task. This history could come from an LLM, or Git commits, or online submissions of (partial or incorrect) solutions of programming tasks. The data is processed and categorized into HCU, HC, CU or C buckets. The user query U is LLM generated.

Cursor-Core: The data generated in (3) above is used to fine-tune Deepseek-Coder, Yi-Coder AI and Qwen2.5-Coder of different sizes. The benchmarks in (2) are used to evaluate.

The evaluation results show that the fine-tuned model perform better than the base models, and other models in that (size) category. The fine-tuned CursorCore models all perform worse than GPT-4o.

**Strengths:**

+ The paper describes a fairly extensive effort that includes benchmark generation using human annotations,
automated data generation, and fine tuning.

+ Evaluation across several models, including several open-source models, across sizes

**Weaknesses:**

- While the effort is impressive, the take-home message remains unclear. The paper starts by emphasizing the need for history (H) -- but inclusion of H almost always leads to worse performance (going from C to HC or going from CU to HCU) -- in fact, that is also true for GPT-4o, which is evidence contrary to the thesis of the paper. The code (C) and user interaction (U) part is already part of most LLM-based programming assistants.

- The abstract also emphasizes conversational framework, but I did not find any conversational interactions here, just the history of the code.

- The presentation is poor because the paper tries to pack too much information in the limited space.

**Questions:**

1. what is the conversational bit in the Assistant-Conversation framework?

2. Lines 288-293 discuss the process of discarding some segments that are not aligned with "user's purpose" -- is there any check in place to ensure that this step does not throws away some important segments? How do you decide if something is aligned with user's purpose when you may not even have the user query (U)?

3. Line 340-343 mention "randomly utilize two powerful open-source LLMs" -- utilize for what?

4. Line 295-296 mention things that are mentioned only there in the main part of the paper. What are the learnings from that part of the paper? Is there a conclusion to be drawn, and if so, then it should be in the main paper, and if not, then adding 2 sentences with a pointer to the Appendix is not very helpful.

---

> ### Author Response · Authors · 2024-11-14
>
> Thanks for your review. Please see our detailed response below:
>
> > This point about history (H) not being incorporated in prediction has been made previously in many papers in software engineering conferences
>
> Our work focuses on proposing a framework that can integrate various types of information, rather than being limited to historical edit records alone. The software engineering community has made significant contributions to utilizing historical edit records, but most of these approaches either rely on prompt engineering techniques or involve modifications to encoder-based or encoder-decoder architectures such as CodeBert or CodeT5, which are not easily applicable to the current LLM training that primarily uses decoder-based architectures. We will update the related work section to reflect these points.
>
> > but there is no mention of releasing the benchmark set in the paper
>
> We release all data collection pipelines, models, and benchmarks. Due to the anonymous review policy, we cannot disclose public code link. We will include the corresponding links in the abstract of the final published version.
>
> > inclusion of H almost always leads to worse performance
>
> This is a normal result because the evaluation samples for different data types are distinct. During the benchmark creation process, in order to better assess the model's ability to utilize historical edits and integrate them with user instructions, we specifically collected data that requires relevant information for inferring user intent. For example, in the case of the H+C and H+C+U categories, these samples were designed to require related information to answer the user query. If a sample only includes data from a single category (such as only C or only U), it might be insufficient to answer the query due to a lack of information. The difficulty of utilizing historical edit information is even greater. Although it can be automatically captured without requiring user input, it reflects user intent in a more ambiguous way, or it may contain irrelevant or noisy records. Therefore, it is harder for the model to make use of this information, which is why the setting with H performs worse than without H.
>
> > The code (C) and user interaction (U) part is already part of most LLM-based programming assistants.
>
> In the current implementation, the components C and U are mixed together in both the model's input and output. Many existing tools handle them through prompt engineering and post-processing techniques to achieve features like instructional editing. However, models like ChatGPT produce outputs in a free-form conversational style. While this format is user-friendly and easy to read, it may hinder automated parsing of the output. For example, to quickly highlight edited sections, LLMs might omit unchanged parts or split the content into multiple snippets, which can pose challenges for applications. To address this, we separate the C and U components in our framework design, making it easier to process the input and output of C.
>
> > I did not find any conversational interactions here / What is the conversational bit in the Assistant-Conversation framework?
>
> We refer to the 'conversation interaction' here just as the interaction between human programmers and AI programming assistants (models). Our Assistant-Conversation framework is proposed as an extension of frameworks similar to ChatGPT's chatbot model. Unlike traditional chatbots, which typically involve only user instructions and model responses, our framework incorporates edit history, current content and user instructions.

---

> > ### Author Response · Authors · 2024-11-14
> >
> > > The paper tries to pack too much information in the limited space
> >
> > Thanks for this suggestion. We will add some transitional sections to further explain the relationship between each module and its real-world applications, as well as the design motivations behind the Assistant-Conversation Framework, to help readers better understand the overall structure. Details are as follows:
> >
> > - Provide a detailed explanation of two examples in the caption of Figure 2, each using different types of information.
> > - Provides additional information on the statistics of H, C, and U in Table 1.
> > - In the caption of Figure 7-8, add analysis of the differences in distribution of different data sources.
> > - Add discussion of other work that utilizes historical information to the related work section.
> > - Add an explanation on how the benchmark data was collected for the targeted evaluation.
> > - Add figures of benchmark examples
> > - Further elaborate on the connection between real-world applications and the motivation behind the design of the Assistant-Conversation Framework in the transition between the Introduction and section 2.
> >
> > > Is there any check in place to ensure that this step does not throws away some important segments? How do you decide if something is aligned with user's purpose when you may not even have the user query (U)?
> >
> > We make judgments using the LLM-as-a-judge method and employ a principle-guided approach to assess whether something aligns with the user's intent. Specific prompts for this can be found in Table 12. While using an LLM to synthesize data cannot guarantee 100% accuracy, it is sufficiently fast and, compared to human annotation, is much more cost-effective. As a result, similar approach has seen some applications in the LLM field, such as RLAIF [1] mentioned in the paper, which uses AI to label human preference data.
> >
> > [1] Constitutional ai: Harmlessness from ai feedback
> >
> > > Line 340-343 mention "randomly utilize two powerful open-source LLMs" -- utilize for what?
> >
> > Utilize for the data collection of training (the data synthesis pipeline outlined in section 4).
> >
> > > Line 295-296 mention things that are mentioned only there in the main part of the paper. What are the learnings from that part of the paper?
> >
> > Our motivation is to align the model using all available information during the programming process. The main information sources we leverage include historical edits, current code, and user instructions. Additionally, other information, such as cursor position and selected areas, are also relevant and can be utilized during programming, which is why we mention them in the main body. However, compared to other types of information, they are not the primary focus, and due to space constraints, we moved the detailed discussion of them to the appendix.
> >
> > We appreciate your review and look forward to your response!

---

> > > ### Comment · Reviewer_zutF · 2024-11-23
> > >
> > > I thank the authors for their detailed response. Based on reading the other reviews, and the authors' response to them, I do have a better appreciation for the work. I agree that there is considerable effort that has gone behind this work. However, I continue to have a hard time evaluating the significance of the contribution. I am not able to definitively point to one thing that is truly a novel contribution and a learning from this work. Observations about the role of code edit history, current code, and user interactions in predicting code transformations have been made before, and so have the observations about gains from fine-tuning and how fine-tuned models require smaller prompts. (I did not fully get the comment about the encoder/decoder-only reference in the response and why it matters here.) The datasets could be a useful contribution, but one would have to see and use the dataset to truly know its value; in any case, the authors claim this is not a benchmark paper. This leads me to keep my evaluation unchanged for the moment.

---

> ### Author Response · Authors · 2024-11-23
>
> Thanks very much for your response!
>
> We will break down and detail the novel contributions of our work as follows, highlighting the innovative aspects:
>
> 1. Consolidate historical edits, current code, and user instructions into a unified format for edit prediction
>
>    We agree with your point that leveraging historical edits has been explored in prior software engineering research. However, some programming assistance scenarios require the simultaneous consideration of historical edits, user instructions, and other contextual information. As we know, previous works have typically focused on a single type of information, such as historical edits or user instructions, without exploring frameworks or models that integrate multiple types of information. Our work is the first to explore it.
>
> 2. Novel Data Collection Pipeline
>
>    We developed an innovative data collection pipeline to automatically gather datasets containing diverse information about the programming process. As reviewer fkg9 mentioned, fine-grained coding history data can be difficult to obtain, this synthetic approach is great for this task. And as we know,, such a method has not been explored in prior work.
>
> 3. Benchmark Contribution
>
>    The benchmark we proposed is another significant contribution. It was made publicly available over a month ago, but due to the double-blind review policy, we are unable to include the corresponding link here. We will add the link in the abstract of the final public version of our paper.
>
> 4. Models
>
>    We release the collection of trained models, which are useful for deployment in applications such as automated editing. As mentioned in the introduction section, existing state-of-the-art open-source Code LLMs are not well-suited for these tasks, and no closed-source APIs currently address these needs. As we know, this is the first fine-tuned Code LLMs specifically trained and released for such tasks.
>
> 5. Design the conversational framework based on the characteristics of modern inference engines for decoder-only LLMs
>
>    Successfully addressing this task requires considering AI/LLM mechanisms (e.g., fine-tuning and inference), systems (e.g., prefix caching and speculative decoding), and software engineering (e.g., specific forms of programming assistance tasks) collectively. For example, tasks like pressing the Tab key for auto-editing demand extremely low latency, requiring us to leverage modern inference systems to optimize speed in the design of conversational framework. As mentioned in Section 2.3, prefix caching is an optimization technique applicable only to decoder-only LLM architectures. We specifically adjusted the conversational framework to accommodate this technique, similar ideas and principles has not been explored in prior work.
>
>    Additionally, some previous works utilized encoder-based methods for localization, these approaches cannot be directly applied to modern decoder-only LLM architectures. Hope this clarifies the point you did not fully get.
>
> This work spans multiple fields and incorporates a significant amount of content. Successfully integrating all these elements is essential to achieving the final application. We use terminology more commonly employed in the AI/LLM domain to align better with ICLR's focus. We fully understand the questions and concerns you raised while reviewing our work, and are deeply grateful for the time and effort you have dedicated to reading and understanding our work.
>
> We sincerely appreciate your review again and look forward to your response!

---

> > ### Author Response · Authors · 2024-11-25
> >
> > Dear Reviewer zutF,
> >
> > We have provided targeted responses based on your comments. Considering the author-reviewer discussion is ending very soon, we would like to follow up to ensure there are no unresolved questions or concerns. If there are any additional points we can address to help the reviewer reconsider their score, please let us know. Thank you again and look forward to your feedback!

---

### Official Review · Reviewer_fkg9 · 2024-11-01

**Soundness:** 3
**Presentation:** 3
**Contribution:** 3
**Rating:** 6
**Confidence:** 4

**Summary:**

This paper presents a dataset synthesis pipeline to generate mixed style programming data to support in-context code generation task. The paper considers a mixture of history steps, current context, user instruction for generation code completion together with rationale as the task. To collect training and evaluation dataset, the authors consider (1) use LLM to generate intermediate steps from an initial program, (2) collect from github commits, and (3) collect coding platform submits. A random mixture approach is taken to generate tasks, with LLM as a judge to compare consistency.

The authors present three models trained on this dataset (with mixture from all sources), and CursorCore models show impressive improvement over baseline models of similar size.

**Strengths:**

This paper's main contribution is the dataset synthesis approach, that leverages AIprogrammer to generate code edit histories from seed code snippets. Then, together with the mixture strategy, APEval includes a diverse set of insertion, editing, generation tasks. Given that fine-grained coding history data can be difficult to obtain, this synthetic approach is great for this task.

The dataset selection process also highlights the benefit of mixing datasets with both chat data and synthetic data. It seems like the gain from github and online coding platform is minimal, which potentially requires additional analysis.

**Weaknesses:**

There are two weakness of the paper:

1. As a benchmark paper, this paper doesn't provide a very convincing evaluation dataset. The authors should provide analysis of the evaluation dataset, justifying (1) why the evaluation set reflects practical editing needs (since AIprogrammer generated data may not completely align with typical programming styles), (2) how does the evaluation set ensures high accuracy, especially with respect to the contextual information (as the authors filtered with the AI judge).

2. The paper doesn't make clear comparison with dataset curated from github/coding submits and synthetic data, despite dataset selection section clearly show that git-commit and online submit made minimal contribution on top of AIProgrammer + Evol-instruct. I suggest the authors provide some insights about their difference qualitatively (e.g., how does the editing styles different among them). If possible, maybe perform cross-evaluation between models trained on these subsets (e.g., train on AIProgrammer, but eval on git data / then train on git data and eval on AIProgrammer data). This might highlight and justify how these datasets should be mixed together.

**Questions:**

As mentioned above, the authors should:
(1) elaborate why eval dataset is convincing, and should be used by future model developers for the general coding assistant benchmark.
(2) compare in detail about data from three sources.

---

> ### Author Response · Authors · 2024-11-14
>
> Thanks for your review. Please see our detailed response below:
>
> > As a benchmark paper, this paper doesn't provide a very convincing evaluation dataset
>
> The reviewer seems to have misunderstood our experimental setup. Our paper is not a benchmark paper, the proposed new benchmark is only part of the work; it encompasses a framework, data collection, training, and benchmarking, among other components. The training dataset we collected and the benchmark used for evaluation are entirely different. The training data was synthesized using LLMs, and during this process, techniques similar to AI Judge were employed to filter the data. The benchmark used for evaluation was manually annotated by human programmers. This benchmark naturally reflects real-world editing needs and is tested using execution-based metrics to ensure precise evaluation of functional correctness. These details are discussed in section 3 and 4 of our paper. The separation between training data construction and benchmark creation is a common practice in the alignment of LLMs.
>
> > The paper doesn't make clear comparison with dataset curated from github/coding submits and synthetic data
>
> We present concrete examples in Figure 2 and provide statistical information from different data sources for further analysis in Figures 5 through 8. During the training of code LLMs, it is not feasible to perform cross-validation on the training data. Currently, evaluation of code models primarily relies on execution-based metrics such as Pass@k to ensure correctness, which requires the sample to be executable and accompanied by accurate test cases. However, the training data does not include corresponding test cases, and some code snippets have highly complex dependencies, making execution nearly impossible. Therefore, for training code LLMs, the approach typically involves training with different data combinations through ablation studies, and comparing results on benchmarks, which is the method used in our paper. And Due to the high cost of training large models, many current works omit comprehensive ablation experiments on the training data in order to accelerate the experimental process. [1] [2]
>
> [1] OctoPack: Instruction Tuning Code Large Language Models, ICLR 2024
>
> [2] Magicoder: Empowering Code Generation with OSS-Instruct, ICML 2024
>
> We appreciate your review and look forward to your response!

---

> > ### Comment · Reviewer_fkg9 · 2024-11-14
> >
> > Thanks for clarification.
> >
> > - Indeed I misunderstood the construction of APEval, I thought it was a subset of collected data, but it turned out to be human annotated data on top of existing benchmarks.
> >
> > - Yes I understand it is not always feasible for run all ablation. I was just a little bit let down that these three sources do not blend as good as my impression from reading the intro (based on Figure 9) and there is not much insights about how these datasets help with each other in practical tasks. But at least they improve the overall performance when they are all blended together.
> >
> > Thanks for the response, I'll update my score to reflect the discussion.

---

> > > ### Author Response · Authors · 2024-11-20
> > >
> > > Thanks for your response.
> > >
> > > It seems that the original score has not been updated yet. Is there anything else you would like to discuss? We would be happy to address any questions or concerns you may have!

---

> ### Author Response · Authors · 2024-11-25
>
> Dear Reviewer fkg9,
>
> Considering the author-reviewer discussion is ending very soon and the updated score you mentioned is not yet reflected in the system (currently still showing Rating: 5 and Confidence: 4), we want to follow up to ask if there is anything else you would like to discuss (or just a minor technical issue with updating the score in the system). Thank you again and look forward to your confirmation!

---

> > ### Comment · Reviewer_fkg9 · 2024-11-27
> > **Updated**
> >
> > Thanks! I have updated scores to reflect my up-to-date understanding.

---

### Official Review · Reviewer_13qq · 2024-11-03

**Soundness:** 3
**Presentation:** 3
**Contribution:** 3
**Rating:** 5
**Confidence:** 3

**Summary:**

The paper presents CursorCore, an AI-powered programming assistant that improves programming support by utilizing multiple sources of information, including code change history, current code and user intention, throughout the coding process. CursorCore introduces a new framework, Assistant-Conversation, and establishes a benchmark, APEval, to assess its effectiveness. The work also develops the Programming-Instruct pipeline for training data synthesis. Evaluation results in HumanEval benchmark shows the effectiveness of CursorCore.

**Strengths:**

- The paper addresses a critical limitation of current code benchmarks by focusing on continuous code editing, aligning its benchmarks more closely with real-world development scenarios.
- The paper introduces Programming-Instruct, a data synthesis pipeline that generates a substantial dataset of 219K samples from sources like GitHub. Experimental results demonstrate that this dataset effectively supports supervised fine-tuning, making it a valuable resource for training code assistance models.
- The description of the Assistant-Conversation framework is clear and comprehensive.

**Weaknesses:**

- The paper divides the HumanEval benchmark into four settings, each with 41 examples, which limits direct comparison across these settings. This approach uses different data for each setting, making it difficult to assess their relative effectiveness. For instance, settings with code history (H) might not be particularly relevant for a task like HumanEval’s code generation, raising questions about its utility in this context.
- The evaluation relies heavily on code generation tasks, which may not fully capture the utility of incorporating code change history information (H) as input. In Table 4, the setting with H seems perform worse than without H. Expanding to additional tasks beyond code generation could better assess the contribution of code history information and more accurately reflect the value of the proposed framework across a broader range of code assistant applications.
- The experimental results analysis is insufficient. In Table 4’s 6B+ model comparison, CursorCore underperforms relative to its base models in specific settings, such as DS-Coder’s C+U, Yi-Coder’s C, and Qwen-Coder’s H+C and C+U settings. This raises questions about the factors driving these inconsistencies, suggesting that the paper’s analysis of model performance across settings could be more thorough to address the causes of these variances.

**Questions:**

- How do you determine the appropriate timing for recording code changes? In real-world scenarios, code changes are often continuous; for example, a modification to a single code block may consist of multiple edits across different lines, and each line may undergo multiple token-level edits. How are boundaries between different history records (H1, H2, etc.) defined?
- How does the framework address the issue of long inputs? With a large number of code changes, the input text can become very lengthy. What strategies are used to manage or reduce input length while preserving relevant information?
- How is the code change history constructed for APEval during manual annotation? Given that HumanEval’s inputs are only function declarations, how do annotators simulate the evolution of a function declaration through code changes to build realistic histories?

---

> ### Author Response · Authors · 2024-11-14
>
> Thanks for your review. Please see our detailed response below:
>
> > Questions about our benchmark
>
> We do not rely solely on code generation tasks; instead, we cover various code editing scenarios. We only use the testcases and function names from the HumanEval (+) benchmark, without utilizing any other content. The historical information, current code, and user instructions are provided by annotators based on the specified function functionality. For instance, our benchmark accommodates scenarios that require edits and historical context. We illustrate this with two simple examples to facilitate quick understanding.
>
> Example 1:
>
> ```python
> # Current
> def has_close_elements(n, t):
>     for i in range(len(n - 1)):
>         for j in range(i + 1, len(n)):
>             if n[i] - n[j] < t or n[j] - n[i] < t:
> ```
>
> This code check if in given list of numbers, are any two numbers closer to each other than given threshold. The current conditional logic in this code has an issue. We should check whether the absolute difference between the two values is less than `t`. Therefore, the model needs to recognize this and, in addition to generating the remaining code, also edit the erroneous code accordingly.
>
> Example 2:
>
> ```python
> # History 1
> def incr_list(l: list):
>     return [x++ for x in l]
>
> # Current
> def incr_list(l: list):
> ```
>
> This example corresponds to a situation where a programmer writes an incorrect piece of code and then deletes it to add the correct version. However, if we only retain the current code, the model cannot infer the specific purpose of this function. For instance, when passing the current code to ChatGPT, it would only respond with something like this:
>
> ```python
> def incr_list(l: list):
>     # ... existing code ...
>     {{ edit_1 }}
>     # ... existing code ...
> ```
>
> While by incorporating historical editing information, the model can infer the user's intent and make accurate code edits. In our benchmark, there are more complex scenarios where the model needs to combine various information to align with the programmer's intent.
>
> During the process of creating the benchmark, in order to better evaluate the model's ability to utilize historical edits and integrate this information with user instructions, we collected samples for the H+C and H+C+U types that required the use of relevant historical information to accurately infer user intent. If a sample contained only a single category of data (such as only C or only U), it might be impossible to provide an adequate answer due to a lack of sufficient information. Leveraging historical edit information is particularly challenging, as it can be automatically captured without user input, but often reflects user intent in a more ambiguous manner or includes irrelevant or distracting records. Consequently, models find it more difficult to utilize this information effectively, and settings that include H perform worse than those without H.
>
> > The experimental results analysis is insufficient.
>
> The reviewer may have misunderstood our experimental setup, and we would like to clarify. Our training is based on base models because fine-tuning LLMs is generally more effective when done on base models rather than on instruction models. The case mentioned by the reviewer pertains to specific categories in our 6B+ model comparison, where our model was compared against the official instruction-tuned versions, not with the base models. Our model significantly outperforms the base models across all categories (average 22.6%) and shows notable improvements in overall performance compared to the corresponding instruction models (average 9.7%), even though our training data is much smaller than the official instruction versions. Additionally, prior models required long prompts to handle this task, whereas CursorCore does not, which is highly beneficial for practical applications.

---

> > ### Author Response · Authors · 2024-11-14
> >
> > > How do you determine the appropriate timing for recording code changes?
> >
> > The appropriate timing for recording code changes is crucial. In the specific system implementation, we treat it as a problem of segmenting edits at different granularities. Code edits can occur at different levels of granularity. For instance, a coarse-grained approach records only large changes, while a fine-grained approach tracks even the modification of a single character. Our design for it is that the model should be capable of handling code changes at any level of granularity, this is why it is not mentioned in the main body. In our data synthesis process, we do not impose any restrictions on the granularity of edits. Instead, we generate modifications at various levels of granularity to support this flexibility.
> >
> > In deploying the model for practical applications, we use simple functions to automatically capture the edit history. How this history is captured can be determined by either the editor or the user. For example, we have two simple implementations as follows:
> >
> > ```python
> > def if_continuous_modify(code1, code2, code3):
> >     dist1 = Levenshtein.distance(code1, code2)
> >     dist2 = Levenshtein.distance(code2, code3)
> >     dist3 = Levenshtein.distance(code1, code3)
> >
> >     if dist3 == dist1 + dist2:
> >         return True
> >     else:
> >         return False
> >
> > def blockwise_if_continuous_modify(code1, code2, code3):
> >     dist1 = Levenshtein.distance(code1, code2)
> >     dist2 = Levenshtein.distance(code2, code3)
> >     dist3 = Levenshtein.distance(code1, code3)
> >
> >     past_diff_blocks = generate_diff_blocks(code1, code2)
> >     new_diff_blocks = generate_diff_blocks(code1, code3)
> >
> >     if dist3 == dist1 + dist2 and len(past_diff_blocks) == len(new_diff_blocks):
> >         return True
> >     else:
> >         return False
> > ```
> >
> > They use different granularities and methods to determine the boundaries between different historical records. The first approach uses Levenshtein edit distance to determine the boundary, where a discontinuity in the edit distance marks the start of a new historical edit record. The second approach considers both the Levenshtein edit distance and the number of diff blocks. Editors or users can decide how to capture historical edits based on practical usage.
> >
> > > How does the framework address the issue of long inputs?
> >
> > We mentioned this in Section 2.3 and discussed it further in Appendix F. Similar to many conversation retrieval techniques used for chatbots in the past, we can compress/retrieve historical edits to address this issue. In this work, we explore a simple approach, the sliding window. When the number of historical edit records exceeds a predefined threshold, the model automatically discards the oldest entries. It is straightforward but yields great results.
> >
> > > How is the code change history constructed for APEval during manual annotation?
> >
> > We inform the annotators about the function's entry point and its purpose, and allow them to send instructions to the AI programming assistant at appropriate moments. We then use screen recording tools to capture the annotators' process of wrtining this function. Afterward, we manually analyze the recordings to construct our benchmark.
> >
> > We will provide a more detailed explanation of the relevant content in the benchmark section of the paper, along with specific examples, to help readers better understand. Details are as follows:
> >
> > - Add an explanation on how the benchmark data was collected for the targeted evaluation.
> > - Add figures of benchmark examples
> >
> > We appreciate your review and look forward to your response!

---

> > > ### Comment · Reviewer_13qq · 2024-11-21
> > >
> > > Thanks for the detailed response.
> > >
> > > Some of my questions are addressed. However, the question in Weakness 1 is still unanswered. Dividing the HumanEval benchmark into four settings, each with different test data, limits direct comparison across these settings. Why not test each of the four settings on the full test set?

---

> > > > ### Author Response · Authors · 2024-11-21
> > > >
> > > > Thanks very much for your response! We are delighted to further discuss the settings of our benchmark.
> > > >
> > > > We believe your question is why we did not construct the benchmark in the following way (If we have misunderstood your intent, please correct us):
> > > >
> > > > > For each sample in the original HumanEval benchmark:
> > > > > 1. Uniformly annotate the programming process to include H, C, and U, then split it into four corresponding combinations.
> > > > > 2. Or annotate four different programming processes containing the four types of information combinations.
> > > > >
> > > > > And then test the entire benchmark.
> > > >
> > > > The reviewer’s suggestion of direct comparison across four settings is intuitive; however, there are some practical challenges in the actual benchmark construction process, which prevented it from being used.
> > > >
> > > > There are several reasons why we did not take this approach. One reason, as mentioned in our previous response, is that removing H or U results in insufficient information, making it difficult for the model to predict the programmer's intent. ("If a sample contained only a single category of data (such as only C or only U), it might be impossible to provide an adequate answer due to a lack of sufficient information")
> > > >
> > > > If you are referring to the second annotation approach (annotating four different programming processes for each task), the issue lies in the differences in information (H, C, U) collected from different programming processes. These differences make direct comparisons unfeasible.
> > > >
> > > > Additionally, for some tasks, it is nearly impossible to collect real-world programming processes containing specific types of information (e.g., those involving H or U). By using human-annotated programming processes rather than slicing parts of the training data or using synthetic data, we aimed to reflect the real-world distribution of programming processes. So we did not impose restrictions on the programmers’ coding habits and only informed them that they could consult the AI programming assistant when necessary.
> > > >
> > > > In our benchmark collection process, we initially annotated one programming process for each task. For some tasks, the annotators consulted the programming assistant; for others, they did not. Similarly, some tasks involved complex editing histories, while others did not. Upon reviewing the data, we found that for certain tasks, it was nearly impossible to collect realistic programming processes containing specific types of information (e.g., H or U). For example:
> > > >
> > > > - Some tasks are straightforward and can be completed with just a few lines of code. Programmers who have undergone basic training can write these solutions quickly without needing to consult an assistant or repeatedly revise their code.
> > > > - Conversely, some tasks may involve calling specific libraries or algorithms that most annotators are unfamiliar with, leading them to rely on the programming assistant.
> > > >
> > > > It would be unrealistic and counterproductive to instruct annotators to "always consult the AI" or "edit your code repeatedly," as this would deviate from real-world scenarios and undermine our intention to use human-annotated data.
> > > >
> > > > Considering these reasons, we did not collect programming traces for the entire test set. While we still hope that the number of samples of four different combinations is at least balanced. At this stage, the number of samples for combinations involving all four data types was relatively similar. So we asked annotators to label additional programming process traces for combinations with fewer samples and collected the corresponding traces. Meanwhile, for combinations with slightly more samples, we discarded some of their traces. Through this process, we established our final benchmark.
> > > >
> > > > Your deep understanding of our work and the detailed questions you’ve raised have been incredibly helpful. We sincerely appreciate your review again and look forward to your response!

---

> > > > > ### Author Response · Authors · 2024-11-25
> > > > >
> > > > > Dear Reviewer 13qq,
> > > > >
> > > > > We have provided targeted responses based on your comments. Considering the author-reviewer discussion is ending very soon, we would like to follow up to ensure there are no unresolved questions or concerns. If there are any additional points we can address to help the reviewer reconsider their score, please let us know. Thank you again and look forward to your feedback!

---

> > > > > > ### Author Response · Authors · 2024-11-27
> > > > > >
> > > > > > Dear Reviewer 13qq,
> > > > > >
> > > > > > We have provided targeted responses based on your comments. As the author-reviewer discussion period is coming to a close, we would like to follow up to ensure there are no unresolved questions or concerns. If there are any additional points we can address to help the reviewer reconsider their score, please let us know. Thank you again and look forward to your feedback!

---

> > > > > > > ### Author Response · Authors · 2024-11-29
> > > > > > >
> > > > > > > Dear Reviewer 13qq,
> > > > > > >
> > > > > > > We have provided targeted responses based on your comments, and uploaded a revised PDF, which includes points discussed with the four reviewers. Revisions are highlighted in blue for clarity. As the author-reviewer discussion period is coming to a close, we would like to follow up to ensure there are no unresolved questions or concerns. If there are any additional points we can address to help the reviewer reconsider their score, please let us know. Thank you again and look forward to your feedback!

---

> > > > > > > > ### Author Response · Authors · 2024-12-02
> > > > > > > >
> > > > > > > > Dear Reviewer 13qq,
> > > > > > > >
> > > > > > > > We have provided targeted responses based on your comments, and uploaded a revised PDF, which includes points discussed with the four reviewers. Revisions are highlighted in blue for clarity. As the author-reviewer discussion period is only one day away from closing, we would like to follow up to ensure there are no unresolved questions or concerns. If there are any additional points we can address to help the reviewer reconsider their score, please let us know. Thank you again and look forward to your feedback!

---

> > > > > > > > > ### Comment · Reviewer_13qq · 2024-12-02
> > > > > > > > >
> > > > > > > > > Thanks for the detailed response. However, I would like to keep my score.

---

### Official Review · Reviewer_jgRp · 2024-11-09

**Soundness:** 4
**Presentation:** 3
**Contribution:** 4
**Rating:** 8
**Confidence:** 4

**Summary:**

The paper proposes:
(1) the Assistant-Conversation Framework. This combines multiple components like System (priming to the tasks), History (past edits and changes made in code), Current (The current code being processed), User (instructions for the task) and Assistant (the responses from the model).
(2) APEval: A benchmark for holistic programming assistant evaluation.
(3) CursorCore: A model for assisted programming tasks.

**Strengths:**

- Current agents lack history information and are very specific to the current context which is the key challenge this paper tries to solve.
- Real user scenarios are a more complex framework of interactions similar to the one proposed.
- The benchmark created evaluates information use from different context sources.

**Weaknesses:**

- The improvements with using CursorCode is not clear from the performance differential.
- The conversational framework needs to be motivated better as its hard to understand why this system was chosen. Also, the paper tries to cover a lot of things which makes it hard to focus on the core problem.

**Questions:**

Formatting:
- Caption for figure 2 can be improved.
- Table 1 caption should explain the data.
- Figure 7 and 8 are hard to parse.

Writing:
Related works should talk about other tasks in AI for code that look at the history of changes like OverWatch.

---

> ### Author Response · Authors · 2024-11-14
>
> Thanks for your review. Please see our detailed response below:
>
> > The improvements with using CursorCore is not clear
>
> CursorCore demonstrates significantly improved performance compared to the base model (average 22.6%). It delivers the best results while requiring far less training data than official instruction-tuned versions (average 9.7%), as shown in Table 4. Additionally, previous models needed lengthy prompts to handle this task, whereas CursorCore does not—an important advantage in practical applications.
>
> > The conversational framework needs be motivated better as its hard to understand why this system was chosen
>
> Thanks for the reviewer’s suggestions. The Introduction section has explained the challenges of the current model in practical applications and the necessary directions for improvement. We will further elaborate on the connection between these issues and real-world applications, as well as the motivation behind the design of the Assistant-Conversation Framework, in the transition between the Introduction and section 2. This will help readers gain a clearer understanding.
>
> > Formatting & Writing
>
> We will update the relevant captions and the related work section to improve the overall presentation of the paper. We will make the following changes:
>
> - Provide a detailed explanation of two examples in the caption of Figure 2, each using different types of information.
> - Provides additional information on the statistics of H, C, and U in Table 1.
> - In the caption of Figure 7-8, add analysis of the differences in distribution of different data sources.
> - Add discussion of other work that utilizes historical information to the related work section.
>
> We appreciate your review and look forward to your response!

---

> > ### Author Response · Authors · 2024-11-25
> >
> > Dear Reviewer jgRp,
> >
> > We have provided targeted responses based on your comments. Considering the author-reviewer discussion is ending very soon, we would like to follow up to ensure there are no unresolved questions or concerns. If there are any additional points we can address, please let us know. Thank you again and look forward to your feedback!

---

> > > ### Comment · Reviewer_jgRp · 2024-11-25
> > > **Thanks for the rebuttal**
> > >
> > > Thanks for the detailed response. I got all my questions answered.

---

### Author Response · Authors · 2024-11-20

Dear Reviewers,

Thanks for your thoughtful reviews! We are excited that the reviewers recognized the strength of our paper:

- Real user scenarios are a more complex framework of interactions similar to the one proposed. (jgRp)
- The description of the Assistant-Conversation framework is clear and comprehensive. (13qq)
- Data synthetic approach is great for this task. (fkg9)
- The paper describes a fairly extensive effort that includes benchmark generation using human annotations, automated data generation, and fine tuning. (zutF)

We thank the reviewers for their feedback, and we’re happy to answer any further questions!

---

> ### Author Response · Authors · 2024-12-03
>
> We sincerely thank the reviewers for their insightful and positive feedback, and their time during the review process!
>
> In response to the reviews and to provide further clarification, we have made the following revisions to our paper. Revisions are highlighted in blue for clarity.:
>
> - Further elaborate on the connection between real-world applications and the motivation behind the design of the Assistant-Conversation Framework in the transition between the Introduction and section 2. (jgRp, zutF)
> - Provide a detailed explanation of two examples in the caption of Figure 2, each using different types of information. (jgRp, zutF)
> - Provides additional information on the statistics of H, C, and U in Table 1. (jgRp, zutF)
> - Add analysis of the differences in distribution of different data sources. (jgRp)
> - Add discussion of other work that utilizes historical information to the related work section. (jgRp, zutF)
> - Add an explanation on how the benchmark data was collected for the targeted evaluation. (13qq, zutF)
> - Add figures of benchmark examples. (13qq, zutF)
>
> We hope these clarifications and updates address the reviewers' concerns. Thanks for your kind consideration!

---

### Meta-Review · Area_Chair_UXnw · 2024-12-21

**Metareview:**

This paper formulates the task of LLM-assisted in-IDE programming, where a model generates code given a system instruction (S), history code changes (H), current programmatic context (C), as well as a user prompt (U). The authors also built an execution-based evaluation benchmark APEval by repurposing coding tasks in the HumanEval dataset, consisting of 164 programming assistance problems with varying types of contextual information (e.g., with or without history code changes H). To adapt LLMs to this new task, the authors also proposed several approaches to generate synthetic instruction tuning data following the same format, using resources like Github commits, submissions to online judges like LeetCode, or LLM-curated coding tasks. Finally, the authors fine-tuned several LLMs using the synthetic data and showed improved performance on APEval compared to prompting existing models.

**Strengths:**

* Compared to other formulations of code generation or editing tasks, the proposed setup of generating code given historical code changes along with current code context and user instructions is more aligned with real-world in-IDE coding assist scenarios (jgRp, 13qq).
The proposed benchmark, APEval and the synthetic data generation pipeline consider different types of contextual information (jgRp, fkg9). The proposed method to generate code edit history is particularly valuable given that “coding history data can be difficult to obtain” (fkg9). Overall, this is “a fairly extensive effort” (zutF).

* The resulting instruction tuning dataset is “substantial” in size (13qq), and the positive experimental results suggested “it a valuable resource for training code assistance models” (13qq). Evaluation also seems comprehensive, with results “across several models, including several open-source models, across sizes” (zutF)

**Weaknesses:**

While this paper presents positive results and involves a fairly extensive amount of effort, there are still several opening issues yet to be addressed:

1. **Limitation of the Benchmark** (R-13qq) There are only 41 evaluation examples covering each context setting (H+C+U **;** H+C **;** C+U **;** C only), and different underlying HumanEval problems are used in these four settings. This makes it difficult to compare scores across different settings and assess their relative effectiveness. While the authors provided convincing reasons of creating problems with different contextual settings from different HumanEval problems, I found the size of each split still very small (41 tasks), making it difficult to interpret the statistical significance of the mainline results in Table 4, where each pass@1 metrics are estimated only using a small group of 41 tasks.

2. **Limited study on the value of training data from different sources** (R-fkg9) The dataset selection section “clearly showed that git-commit and online judge submissions made minimal contribution on top of AIProgrammer + Evol-instruct”, and “there is not much insight about how these datasets help with each other in practical tasks”. Additionally, there is a potential mismatch between the distribution of problems in the APEval benchmark and those in the training data. In particular, HumanEval problems only consider simple algorithmic tasks using built-in Python functions, and may not reflect the benefit of training data that feature a broader spectrum of more realistic use cases and domains, such as data derived from arbitrary github commits.

3. **Overall Significance of the Contribution** (R-zutF) “Observations about the role of code edit history, current code, and user interactions in predicting code transformations have been made before, and so have the observations about gains from fine-tuning and how fine-tuned models require smaller prompts.”

4. **Potential Misuse of Terms** (zutF) The usage of “conversational framework” in the paper is potentially misleading, as this term has specific meanings in NLP, in particular dialogue research. There is nothing that is "conversational" in the APEval dataset and the instruction following data (e.g., there is only one turn of user interaction, i.e., there is at most one user prompt in each problem).

5. **Issues with Writing** There is a lot of room for improvement in the overall writing and presentation. To name a few:

* “Online submit”: consider renaming to "submissions to online coding platforms”

* Sec 3.1: “Table 1: APEval Statistics. Present …” -> “Table 1: APEval Statistics and breakdown of tasks by context type”.

* L139: Consider using a more precise noun to refer to $C$, for example, “code context”, instead of using “current”. Similarly, consider using “User Prompt” to refer to $U$ and "Assistant Output” for $A$.

Given these remaining issues, this paper could benefit from another round of revision. Hence the recommendation is Rejection.

**Additional Comments On Reviewer Discussion:**

Please see the above review.

---

### Decision · Program_Chairs · 2025-01-22

Reject